# Low-Diversity Microbiota in Apical Periodontitis and High Blood Pressure Are Signatures of the Severity of Apical Lesions in Humans

**DOI:** 10.3390/ijms24021589

**Published:** 2023-01-13

**Authors:** Matthieu Minty, Sylvie Lê, Thibault Canceill, Charlotte Thomas, Vincent Azalbert, Pascale Loubieres, Jiuwen Sun, Jonathan Sillam, François Terce, Florence Servant, Alain Roulet, Céline Ribiere, Michel Ardouin, Jean-Philippe Mallet, Rémy Burcelin, Franck Diemer, Marie Georgelin-Gurgel, Vincent Blasco-Baque

**Affiliations:** 1Département Dentaire, Université Paul Sabatier III (UPS), 3 Chemin des Maraîchers, CEDEX 9, 31062 Toulouse, France; 2Service d’Odontologie Toulouse, CHU Toulouse, 3 Chemin des Maraîchers, CEDEX 9, 31062 Toulouse, France; 3UMR1297 Inserm, Team InCOMM/Intestine ClinicOmics Metabolism & Microbiota, Institut des Maladies Métaboliques et Cardiovasculaires (I2MC), Université Paul Sabatier, 1 Avenue Jean Poulhes, 31432 Toulouse, France; 4Vaiomer, 516 Rue Pierre et Marie Curie, 31670 Labège, France

**Keywords:** apical periodontitis, oral microbiome, oral diseases, high blood pressure, periapical index, granuloma

## Abstract

(1) Background: In developed countries, the prevalence of apical periodontitis (AP) varies from 20% to 50% for reasons that could be associated with the apical periodontitis microbiota ecology. (2) Methods: We performed a clinical study in the Odontology department of Toulouse hospital in France, to sequence the 16S rRNA gene of AP microbiota and collect clinical parameters from 94 patients. Forty-four patients were characterized with a PAI (periapical index of AP severity) score lower or equal to 3, while the others had superior scores (n = 50). (3) Results: The low diversity of granuloma microbiota is associated with the highest severity (PAI = 5) of periapical lesions (Odds Ratio 4.592, IC 95% [1.6329; 14.0728]; *p* = 0.001; notably, a lower relative abundance of Burkholderiaceae and a higher relative abundance of *Pseudomonas* and *Prevotella*). We also identified that high blood pressure (HBP) is associated with the increase in PAI scores. (4) Conclusions: Our data show that a low diversity of bacterial ecology of the AP is associated with severe PAI scores, suggesting a causal mechanism. Furthermore, a second risk factor was blood pressure associated with the severity of apical periodontitis.

## 1. Introduction

Apical periodontitis (AP) is a bacterial infection of the endodontic area of the tooth, with a chronic immunoinflammatory reaction of all the apical supporting tissues of the tooth (bone and desmodontium) [1]. Epidemiological studies in developed countries show that the prevalence of AP varies from 20% to 50% in the general population [2,3]. It is considered by the WHO (World Health Organization) as an infectious pandemic. The histopathological aspect of chronic AP is dominated by the presence of macrophages, mononuclear cells, lymphocytes and plasma cell types [4], which is suggestive of the importance of endodontic bacteria colonization to the periapical area [5]. The relationship between the severity of the AP and the inflammatory reaction could be linked to a specific microbial ecology. The severity of the AP is characterized by alveolar bone loss and immune reaction. As previously described, alveolar bone loss is associated with Gram-negative bacteria [6,7]. The cellular mechanisms involve specific immune reactions, where the pathogen-associated molecular pattern (PAMP)—such as LPS from specific bacteria—triggers the corresponding Toll-like receptor 4 (TLR4) to initiate the inflammatory process, leading to bone loss [8]. A major question is the specific role of endodontic microbiota in the development of AP. In clinical practice, it is usual to classify different types of AP using the periapical index (PAI: a score of the severity of the lesion) following the severity of lesions [9,10]. Hence, it is of major importance to identify molecular mechanisms that could explain or be responsible for the severity of lesions identified by PAI scores. Such a mechanism could be related to the translocation of specific endodontic bacteria to the peri-apex, thereby driving the immune reaction involved in the severity of lesions [11]. 

We hypothesized that the translocation of bacteria from the oral microbiota to inside the tooth, then to the peri-apex area and further to systemic circulation, could induce inflammation and contribute to the severity of the granuloma [6]. Here, we decipher the potential mechanisms of this phenomenon. We therefore hypothesize that the virulence of endodontic microbiota and its diversity are major regulators of severity of AP, as suggested for chronic periapical lesions [12]. In addition, high blood pressure could be associated with the severity of AP [13,14]. In fact, according to the literature, many studies have shown that individuals with endodontic pathologies associated with oral dysbiosis [15] may accumulate additional risk factors for high blood pressure or other cardiovascular diseases [16].

The principal aim of this study was to describe, for the first time, the composition of granuloma microbiota, the microbial ecology of AP, following the severity of AP, as well as to find some association, bacterial or clinical, with the severity of AP. To answer this question, we set up a multicentric cross-sectional study with a cohort of 94 patients, and AP material was taken after endodontic surgery to realize the bacterial 16S rRNA gene sequencing.

Our results show that a specific AP microbiota distinguishes the severity of AP, and the diversity of AP microbiotas was associated with impaired periapical lesions. As per previous reports, we confirm that high blood pressure is a frequent comorbidity in patients with high PAI scores [17].

## 2. Results

A total of 94 patients were included in this study: 44 had a lesion with a PAI score lower or equal to 3 and 50 had a lesion with PAI higher than 3.

### 2.1. Description of the General and Clinical Characteristics of the Granuloma and Identification of a Risk Factor “High Blood Pressure”

#### 2.1.1. All Subjects

The clinical characteristics of the participants are presented in Table 1. The mean age was 54.53 years old (±14.22), with a weight of 71.39 kg (±13.89) and height of 169.67 cm (±13.22). All of the participants had a quality-of-life score above 6, ensuring their future comparability, and a stress score (scale 0 to 10) of about 4.68 (±2.68).

The oral health status of the participants was analyzed: the DMF index (decay, missing, filled teeth) of the global cohort was 14.63 (±5.38), the number of dental brushings/day was 2.01 (±0.59) and the average PAI score was 3.64 (±1.09).

#### 2.1.2. Epidemiologic Parameters of the Two Groups: PAI ≤ 3 and PAI > 3

The patients were also characterized according to the periapical index (PAI), a severity-measuring scale for periapical lesions (Table 1). The average age was similar for the two groups, PAI ≤ 3 and PAI > 3 (55.40 ± 14.48 vs. 53.76 ± 13.54, respectively; *p* = 0.36), as well as the weight, height and level of stress, with no statistically significant difference observed for these values (stress scores were 4.71 ± 2.85 vs. 4.56 ± 2.52, respectively; *p* = 0.55). Neither the general characteristics of the patients nor their oral health status were significantly different between the two groups. The DMF index (decay, missing, filled) of the global cohort was not significantly different between the two groups, PAI ≤ 3 and PAI > 3 (15.25 ± 5.83 vs. 14.10 ± 4.95, respectively; *p* = 0.31), which could be explained by the similar number of brushings per day (2.043 ± 0.52 vs. 1.98 ± 0.65, *p* = 0.60 for PAI ≤ 3 and PAI > 3, respectively, Table 1).

We explored the potential link between the severity of the lesion and general health, in order to identify potential clinical factors associated with the PAI score. Different clinical parameters were first measured, then subsequently subjected to principal component analysis (PCA), as shown in Figure 1. The PCA aimed to identify clinical risk factors potentially associated with the severity of the disease, suggesting ways in which hypertension and periapical disease may be associated.

The results showed that the three main systemic clinical factors responsible for the dispersion of subjects in the PCA analysis were stress, gender and high blood pressure (HBP). We observed that HBP was the most closely associated with the severity of AP (PAI). We split the cohort into two further groups: subjects with HBP (n = 19) and without HBP (n = 75), in order to identify a link between HBP and the severity of the lesion. To describe the causal mechanism between the severity of apical lesions and clinical parameters, we performed a taxonomic analysis of our samples.

### 2.2. Low Diversity in Granuloma Microbiota Is Associated with the Highest Severity of AP

The AP microbiota were investigated using bacterial 16S rRNA gene sequencing. Taxonomic profiles showed that patients were split into low- (43 patients) or high-diversity microbiota (51 patients). Low diversity in AP microbiota was characterized by a taxonomic profile with only one, two or three taxa representing most of the total bacterial relative abundances (Figure 2A). Alpha-diversity analyses confirmed that bacterial richness (based on observed OTUs and Chao1) and diversity (based on Shannon, Simpson) were significantly lower in the low-diversity profiles (Figure 2B). Within the low-diversity bacterial profiles, patients number 22, 35 and 39 were dominated by one, two and three taxa, respectively (four patients were also included as low diversity– see Materials and Methods section—cohort low and high diversity). Amongst these bacterial taxa, Enterobacteriaceae, Pseudomonadaceae, Enterococcaceae, Fusobacteriaceae and Staphylococcaceae were frequently detected.

The Enterobacteriaceae family represented more than 90% of the total abundance in 2 of 20 patients with a PAI equal to 5, and it was the predominant bacterial family in 23% of subjects with a low-diversity profile (Figure 2A). The Pseudomonadaceae family represented more than 90% of the total abundance in 3 of 20 patients with a PAI equal to 5, and it was the predominant bacterial family in 14% of subjects with a low-diversity profile (Figure 2A). The Staphylococcaceae family represented more than 75% of the total abundance in 5 of 20 patients with a PAI equal to 5, and it was the predominant bacterial family in 21% of subjects with a low-diversity profile. The Enterobacteriaceae, Pseudomonadaceae and Staphylococcaceae families were the main bacterial taxa represented in subjects with a low-diversity profile for AP microbiota. The data suggested that the low-diversity microbiota could contribute to the severity of PAI ≤ 3 groups.

Following the bacterial taxonomic profiles, we split the cohort into two groups, with high (n = 51) or low diversity (n = 43) of granuloma microbiota. Lower diversity bacterial profiles were more frequently associated with severe granuloma. Among the 43 low diversity bacterial profiles identified, 27 were associated with PAI scores above 3. On the contrary, less severe granuloma were linked to higher diversity microbiota, being found in 55% of patients with a PAI score lower or equal to 3 within the higher diversity microbiota group (Figure 2A).

We identified a strong correlation between the PAI score severity and the granuloma microbiota diversity identified by 16S rRNA targeted metagenomics (Figure 1). The low diversity of granuloma microbiota was associated with the highest severity (PAI = 5) of periapical lesions (Odds Ratio 4.592, IC 95% [1.6329; 14.0728]; *p* = 0.001, Appendix A). In the group with low diversity for AP microbiota, we identified 46.5% of patients with a PAI = 5 (the highest severity), whereas 15.6% patients had a PAI=5 in the high-diversity group. From that cohort, initial 16SrRNA target sequencing was carried out, which showed that the index score can classify patients according to their granuloma microbiota. This suggests that a specific microbiota ecology could be responsible or associated with the severity of disease. In Appendix A, describing the clinical difference between low- and high-diversity groups, we showed a significant difference in the PAI score between the low-diversity and high-diversity groups (3.95 ± 1.13 vs. 3.41 ± 0.98, *p* = 0.01).

### 2.3. Subjects with a Periapical Index PAI > 3 Show a Significant Increase in the Abundance of Propionibacterium, Prevotella7, Pseudomonas and Pseudomonadaceae and a Significant Decrease in the Abundance of the Family Burkholderiaceae

Microbial ecology has been shown to be responsible for many local and systemic infections. Here, we aimed to identify for the first time a bacterial community potentially associated with severe PAI score. We performed 16S rRNA gene sequencing. Following the analysis of the low-diversity bacterial profiles (see above), we determined patients with low or high diversity for AP microbiota as described previously. In the PAI ≤ 3 group, we identified 36% of patients with a low diversity for AP microbiota, whereas 54% patients of the PAI > 3 group had a low-diversity microbiotal profile, suggesting a link between the diversity of microbiota AP and the severity of AP. The data showed that among the high-diversity microbiota profiles, differences were observed between the PAI ≤ 3 and PAI > 3 groups. The Burkholderiaceae family 0.90 ± 1, 0.84 vs. 0.16 ± 0.74, respectively; *p* = 0.0012) was significantly more abundant in the lesions with PAI ≤ 3, compared to the lesions with PAI > 3. The Flavobacteriaceae family and the genus *Captnocytophaga* and *Sphingomonas* were also more abundant in the lesions with PAI ≤ 3 (Figure 3B, Appendix A). The Pseudomonodaceae family and the genera *Propionibacterium* (0.00 ± 0.00 vs. 0.41 ± 0.92, *p* < 0.0001) and *Prevotella* were significantly more abundant in the lesions with PAI > 3 (Figure 3A). We also noticed that the Burkholderiaceae family was absent in the group PAI > 3, except in two patients (Figure 3B. We identified a positive correlation between the abundance of Burkholderiaceae and the alpha-diversity following the Shannon index (R^2^ = 0.13, *p* = 0.0003). No significant separation was observed between the two groups on a two-dimension principal coordinates analysis (PCA) based on the beta-diversity indices (permanova: *p* = 0.3158 and permadisp *p* = 0.1074—Appendix A).

### 2.4. High Blood Pressure Is an Aggravating Factor for the Severity of Periapical Lesion Associated with a Decreased Diversity of Granuloma Microbiota

In the cohort of 94 participants, 75 participants were not hypertensive (HBP−) and 19 participants (20%) were diagnosed as hypertensive (HBP+).

Table 2 presents the characteristics of the participants according to the criteria HBP. The average age was similar for the two groups, HBP− and HBP+ (53.66 ± 13.96 vs. 52.5 ± 13.95, respectively; *p* = 0.77). The two groups were also comparable in terms of weight, height and level of stress, with no significant difference observed for these values (stress scores were 4.66 ± 2.59 vs. 4.65 ± 2.65, respectively; *p* = 0.72) (Table 2). Neither the general characteristics of the patients nor their oral health status were significantly different between the two groups. The PAI score was higher in the HBP+ group compared to the HBP− group (4.05±0.96 vs. 3.47± 1.05, *p* = 0.0416). The most severe granulomas, with a PAI score of 5, were more frequently diagnosed among the HBP+ group (n = 8; 42% vs. n = 20; 27% in HBP− group). We identified a potential correlation between the PAI score severity and HBP.

To investigate a potential link between the diagnosis of HBP and oral health, we collected different clinical parameters (Table 2). The DMF index (decay, missing, filled teeth) of the global cohort was not significantly different between the two groups, HBP− and HBP+ 14.74 ± 5.26 vs. 13.56 ±6.03, respectively; *p* = 0.44), which could be explained by the similar number of brushings per day (1.98 ± 0.58 vs. 2.1 ± 0.65, *p* = 0.44 for HBP− and HBP+, respectively). The severity was higher in groups with HBP+; in fact, the PAI index was significantly higher in patients with HBP+ versus HBP−.

As previously mentioned, granuloma microbiota were investigated using bacterial 16S rRNA gene sequencing. Taxonomic profiles showed that patients were split into HBP+ (n = 19) or HBP− (n = 75). In Figure 4A, we can see that patients with low diversity were more important in HBP+ (n = 10; 53%) patients compared to HBP− (n = 33; 44%).

To evaluate the association between granuloma microbiota and HBP among the high-diversity microbiota profiles (n = 51), we performed a taxonomic analysis represented by a LefSE (Figure 4B) of the granuloma microbiota. The MiSeq analysis showed that the Lachnospiraceae family (0.88 ± 1.60 vs. 0.0003 ± 0.001, *p* < 0.0001) was significantly more abundant in patients with HBP+ than HBP−. The families of Muribaculaceae, Ruminococcaceae and Desulfovibrionaceae, and the genus *Blautia* and *Roseburia*, were also more abundant in the patients with HBP. On the other hand, the Actinomycetaceae (0.016 ± 0.46 vs. 1.41 ± 3.62, *p* = 0.0025), Corynebacteriaceae (0.02 ± 0.07 vs. 0.54 ± 1.65, *p* = 0.0035) and Spingobacteriaceae (0.12 ± 0.37 vs. 1.29 ± 3.73, *p* = 0.036) families were more abundant in the patients with HBP−, as were the Eubacteriaceae, Leptotrichiaceae and Neisseriaceae families (Figure 4B and Appendix A).

The alpha diversity of the granuloma microbiota was significantly different (*p* = 0.0255) for the Shannon index between HBP− and HBP+ (Figure 4C and Appendix A). This difference was not found for the beta diversity (permanova, *p* = 0.8725). Interestingly, it appears that there is a wide difference in the types of bacteria in the granuloma of people presenting high blood pressure and those in people without high blood pressure.

## 3. Discussion

Bacterial infections are linked to different associations of bacterial complexes, ranging from very simple mono-infections to complex pluri-infections. In our study, lower diversity bacterial profiles were more frequently associated with severe granuloma.

Our study also shows that some bacterial species are mainly found in small periapical lesions, and that others are mostly found in bigger lesions. The identification of these bacteria opens the way for personalized treatment before endodontic surgery. Indeed, using pre-operative cone-beam computed tomography (CBCT), which provides information on the size of the lesion, the practitioner will be able to anticipate the types of bacteria that are responsible for a patient’s infection. *Propionibacterium*, for example, which is mostly identified here in the lesions with a PAI of more than 3, is known to be implicated in infection complications (periprosthetic joint infections, infectious bronchitis, etc.) [18]. An efficient treatment for these diseases is antibiotic therapy with Levofloxacin (500mg once a day during 10 to 14 days). The prescription of such a treatment may, thus, be efficient in periapical healing.

The identification of the bacteria mostly implicated again highlights the question of the prevention of periapical lesions. Even if the eradication of the whole endodontic biofilm is not yet possible, the validated procedures for performing an endodontic treatment enable the elimination of most of the bacteria. It will now be important to ensure that these protocols have an effect on the bacteria reported in this study. In addition, more than 40% of the treated teeth result in an endodontic lesion, clearly showing the need for additional understanding of the reason for this high prevalence [19]. The severity (PAI) predicts the clinical approach, such as the recourse of endodontic surgery (ES). The first treatment of AP is an endodontic treatment where only 50% of the patients rapidly heal within 12-24 months. In case of the failure of healing, the clinician performs an ES to remove the granuloma from the apical area, the cleaning of the bacteria inside the canal and the apical retro-filling. Identifying the specific signature of AP microbiota could be a key factor to prevent the use of ES. To identify the clinical and microbiota parameters associated with the severity of the lesion, we performed a principal component analysis (PCA) (Figure 1). The PCA showed that high blood pressure is associated with the severity of granuloma. The review of the literature confirms that high blood pressure is the most frequent comorbidity in patients suffering from periapical abscess (24.6%), an advanced form of endodontic infection/inflammation [20]. In addition, many epidemiologic studies indicate an association between periodontal disease and hypertension [21], and our results confirm the literature. The molecular mechanism of the crosstalk between microbiota and the severity of the endodontic lesion is also described in the literature.

TLR4 signaling is known to be the key type of pro-inflammatory signaling in the induction of hypertension and periapical lesions. Knowing that TLR4 is the major receptor for lipopolysaccharide present in Gram-negative bacteria, we can predict the influence of oral microbiota and its dysbiosis on high blood pressure. In past years, it has been shown that the microbiota, a complex and dynamic population of microorganisms found in a number of organs and tissues, plays a symbiotic role in modulating human health [22]. One of the largest and most diverse microbiotas is found in the mouth, at the interface with the outside world. Individual bacterial components of the microbiota can be considered as “biomarkers”, and the analysis of oral microbiota (by DNA sequencing, for example) allows early identification of specific bacteria predictive for local and systemic diseases. Our results show the increased relative abundance of the Burkholderiaceae family in lesions with a PAI ≤ 3. These are Gram-negative aerobic or anaerobic bacteria [23] that are notably involved in the development of melioidosis with *Burkholderia pseudomallei* [24]. This is an infectious disease mainly found in tropical areas and for which it appears that type 2 diabetes and obesity are major risk factors [22]. Cell reactions are altered, especially for macrophages and CD4+ regulatory T-cells that are unable to give an adapted response to the presence of the bacteria [25]. Thus, there is a link between the presence of the Burkholderiaceae family in humans and metabolic diseases which are responsible for maintaining a systemic inflammatory state. However, these species do not seem to be implicated in the development of cardiovascular complications of metabolic diseases. Indeed, Fak et al. [26] did not highlight them among the oral bacteria considered as atherosclerosis “markers”. In 2011, Koren et al. [27] even reported that the wider family of proteobacteria (to which the Burkholderiaceae belong) were found in a higher proportion in the oral microbiota of patients unaffected by atherosclerosis. In our study, we also noted the absence of an excessive proliferation of Burkholderiaceae species in the periapical granuloma of patients suffering from high blood pressure (HBP). This is consistent with results obtained in a bigger study which showed that microbiota in humans and animals presenting HBP are not generally characterized by high levels of Burkholderiaceae [28]. Chronic infections, including periodontal infections, may predispose to cardiovascular disease. In our study, we also found that patients with HBP were more likely to have major granuloma (22% vs. 44% PAI = 5) than patients not showing HBP. For the moment, there are insufficient data to clearly establish if the presence of these species of bacteria is somehow protective against the development of vascular complications of metabolic diseases.

Interestingly, we also found an association between the high severity of the periapical lesion and a low diversity in oral microbiota, an observation that has already been reported in the literature when comparing the bacterial composition of pathologic versus healthy teeth [29]. In fact, we showed that the low diversity of granuloma microbiota is a risk factor for the highest severity (PAI = 5) of periapical lesions. The results suggest that a specific microbiota ecology could be responsible for the severity of AP. To understand the corresponding molecular mechanisms of the virulence of the microbiome to host immune defense crosstalk, it will be interesting to realize a shot gun sequence from the full microbiota of the granuloma (GranulOmics) and the saliva (SalivOmics) from the discovery library. Identifying molecular pathways of the host to microbiome crosstalk associated with the healing index could also be interesting to study. The main deliverable of this study is to define a potential therapeutic strategy to manage the virulence of granuloma microbiota associated with the severity, in order to prevent the recourse of endodontic surgery.

## 4. Materials and Methods

The study reported here was approved by the Commission Ethique du Département de Médecine Générale de Midi Pyrénées in June 2016 and the Minister of Health (France), by the collection number N° DC-2022-5010. The STROBE statement guidelines for reporting were followed.

### 4.1. Study Design and Settings

A cross-sectional study was performed between September 2017 and June 2018 in a public hospital (Hôpitaux de Toulouse, Toulouse, France). This hospital was also the reference center of the study.

### 4.2. Participants

Patients scheduled for endodontic surgery in the operating rooms of the hospital were invited to be included in the study. The only exclusion criterion was the presence of any serious medical condition or health problem that would have contraindicated an appointment for surgery.

The authors attest that each participant received clear and detailed information and provided informed consent for their participation and agreement for an oral examination and biological analyses of their saliva and dental plaque. All subjects included were required to write and sign a consent form after receiving the information on the guidelines and the course of the study.

### 4.3. Data Collection and Variables

Before the endodontic surgery, the size of the lesion was determined using pre-operative CBCT (cone-beam computed tomography) and classified following the periapical index [30]: (PAI scores: 1 = normal periapical structure, 2 = small modifications of the periapical bone structure, 3 = structural modification and mineral loss, 4 = AP with a well-defined radiolucent image, 5 = severe AP with exacerbation. During the surgical procedure, the periapical lesion and the tooth apex were removed. When the patient had given their consent, the surgeon assessed the type of lesion (granuloma or cyst) and determined if it was possible to include the sample in the study. Only granulomatous lesions were preserved, the structural and cyst formation mechanisms being too different.

Efforts to reduce potential confusing biases were made by collecting diverse information, such as the participant’s quality-of-life score (using a 10-point scale), their last visit to the dentist and their level of stress (recorded here again on a 10-point scale). General and oral health characteristics were also collected from participants, i.e., age, weight, height and BMI (Body Mass Index), and HBP. The WHO criteria were used to diagnose the caries status with the decayed, missing, and filled (DMF) index of each participant, focusing only on the decaying origin of missing or filled teeth, in accordance with the WHO guidelines on clinical examination in epidemiological studies (https://cioms.ch/wp-content/uploads/2017/01/International_Ethical_Guidelines_LR.pdf (accessed on 10 January 2023)) (Table 1 and Table 2). The gingival status was assessed using a questionnaire. Three qualified dentists conducted the oral health examinations and the periapical surgery. Prior to the study, the three examiners verified that they were to perform exactly the same procedures. After the surgeries, the samples were frozen and stored at −80 °C until their analysis.

### 4.4. Microbiota Analysis

Total DNA was extracted using a QIAamp Cador Pathogen Mini kit (ref 54,106) from QIAGEN.

#### 4.4.1. Bacterial 16S rRNA Gene Sequencing

Profiling of the oral microbiome was performed at Vaiomer (Labège, FRANCE). Briefly, the V3–V4 hypervariable regions of the 16S rRNA gene were amplified by PCR using universal Vaiomer primers [30] Amplicons (467 bp on the Escherichia coli reference genome) were purified using the magnetic beads CleanNGS for DNA clean-up (CleanNA). All libraries were pooled in the same quantity to generate an equivalent number of raw reads and were sequenced on a MiSeq Illumina platform (2 × 300 bp paired-end MiSeq kit v3, Illumina).

#### 4.4.2. 16S rRNA Gene Sequence Analysis

The targeted metagenomic sequences were analyzed using a bioinformatics pipeline based on ‘find, rapidly, OTUs with Galaxy solution’ (FROGS) guidelines [31]. In brief, after demultiplexing of barcoded Illumina paired reads, single-read sequences were cleaned, and the last 20 and 50 bases of R1 and R2 read, respectively, were trimmed and paired into longer fragments. OTUs were produced with single-linkage clustering. The taxonomic assignment was performed by using BLAST against the SILVA 132 database to determine bacterial profiles from phylum to genus, and when reachable, to species level. The following filters were applied: amplicons with a length of <350 nt or a length of >500 nt were removed, and OTUs with an abundance lower than 0.005% and that appeared less than twice in the entire dataset were removed. Alpha and beta diversity analyses were conducted on the OTU table.

#### 4.4.3. Statistical Analysis

Significant variations in alpha diversity were assessed using the Kruskal–Wallis test or the Wilcoxon rank-sum test. Multidimensional scaling analyses (MDS) were performed on beta diversity distance matrices, and differences between groups were assessed using PERMANOVA and PERMDISP analyses (2000 permutations). LEfSe (linear discriminant analysis effect size) analyses based on non-parametric tests were used to determine significant variations in taxa relative abundance [32].

### 4.5. Data Management and Analysis: Statistical Analyses

Data were blinded to maintain participant confidentiality. The comparisons between the groups were performed with Mann–Whitney Wilcoxon tests for quantitative variables and with Fisher tests for qualitative variables. The bacterial diversity analyses (Chao1 alpha diversity and Bray–Curtis beta diversity) were performed using the Phyloseq v1.14.0 R package. The differential taxa analyses were conducted using the linear discriminant analysis effect size tool, LEfSe, using default parameters (alpha parameter significance threshold set to 0.05 and the logarithmic LDA score cut-off set to 2.0). Unpaired Mann–Whitney tests were performed using GraphPad Prism (GraphPad Software. San Diego. CA). A PERMANOVA test was performed with 2000 permutations to assess the statistical significance of the difference between groups in the beta diversity PCoA analyses.

### 4.6. Cohort Low and High Diversity

Samples with the 3 most abundant OTUs (operational taxonomic units) forming at least 75% of their profile were classified as “Low diversity”. Additional manual curation was performed to also include 4 samples (10, 16, 28 and 49) dominated by bacteria known as opportunistic pathogens (Tannerella forsythia, Pseudomonas aeruginosa, Fusobacterium nucleatum, Staphylococcus epidermidis), which were also considered as “Low diversity”.

## 5. Conclusions

We identified a strong correlation between the highest PAI scores severity and the low granuloma microbiotal diversity (Odds Ratio 4.592, IC 95% [1.6329; 14.0728] *p* = 0.001). PAI scores superior to 3 were associated with lower granuloma microbiotal diversity, a lower relative abundance of Burkholderiaceae and a higher relative abundance of Pseudomonadaceae and Prevotellaceae. We further identified high blood pressure (HBP) as a worsening factor for severe (PAI = 5) apical periodontitis. Our data suggest a correlation between high blood pressure and a high severity of periapical lesions. The novel identification of the bacteria implicated in the development of AP opens the way for a personalized therapy for the treatment and prevention of these lesions. Although the growth of the granuloma does not solely depend on the signature of bacterial microbiota, the potential identification of specific bacteria and systemic factors such as HBP enables new clinical development for dentistry practices. The potential identification of specific antibacterial for treating different types of AP confers clinical relevance on our study.

## Figures and Tables

**Figure 1 ijms-24-01589-f001:**
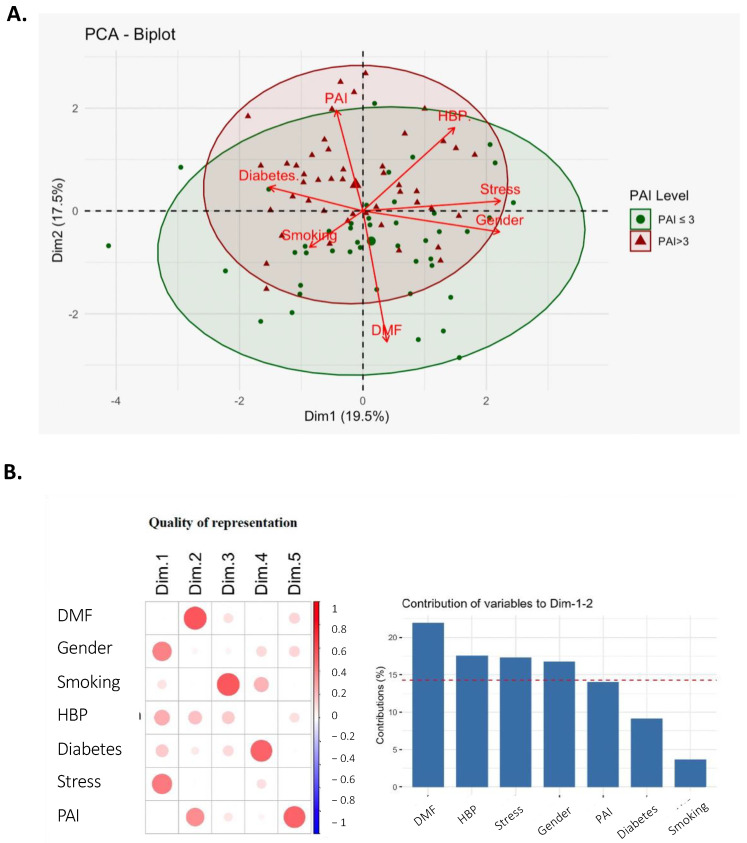
A principal component analysis (PCA) was generated predicting that the main systemic clinical factor responsible for the dispersion of subjects was high blood pressure (HBP). (**A**) PCA between the severity of apical periodontitis (AP) (following the PAI scores), and oral and systemic clinical parameters. (**B**) The statistical analysis for the contribution of each oral and systemic parameter for each dimension.

**Figure 2 ijms-24-01589-f002:**
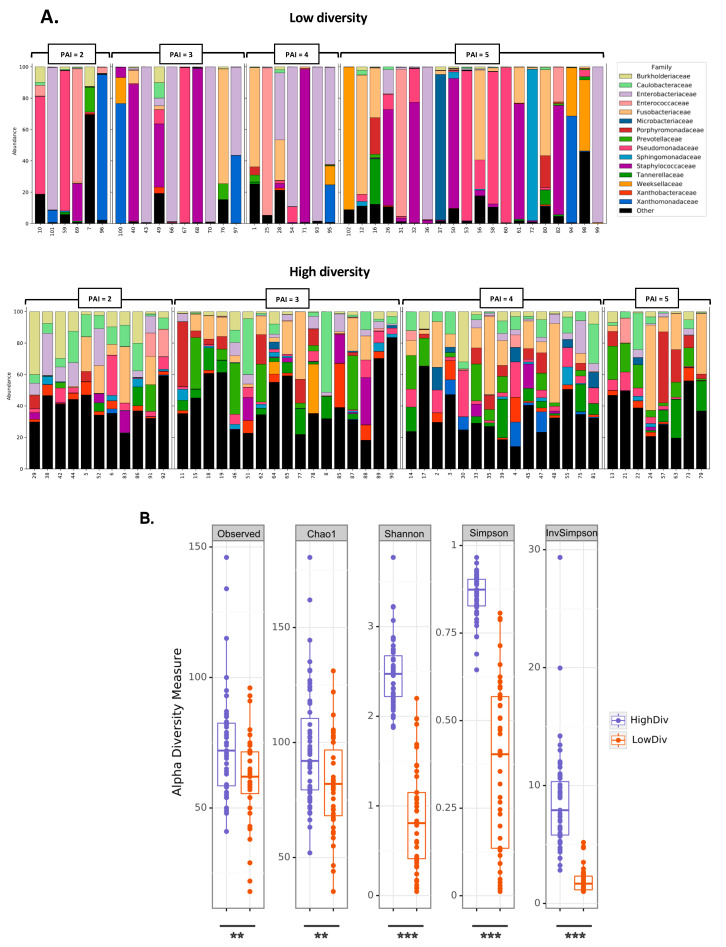
Microbiota profiles from granuloma split into low- (LowDiv, n = 43) and high-diversity (HighDiv, n = 51) profiles. (**A**) Granuloma microbiota composition at family level for low (upper panel) and high (lower panel) diversity. Only the top 15 most abundant families are displayed; the remaining families are aggregated in the “other” category. (**B**) Alpha diversity analyses at OTU level confirm low- and high-diversity microbiome profiles in granuloma. Boxplots show the distribution of bacterial richness (observed OTUs and diversity (Shannon and Simpson) values of granuloma microbiota between low- and high-diversity profiles. Statistically significant differences were determined using a Wilcoxon test (*** *p* value ≤ 0.001, ** *p* value ≤ 0.01).

**Figure 3 ijms-24-01589-f003:**
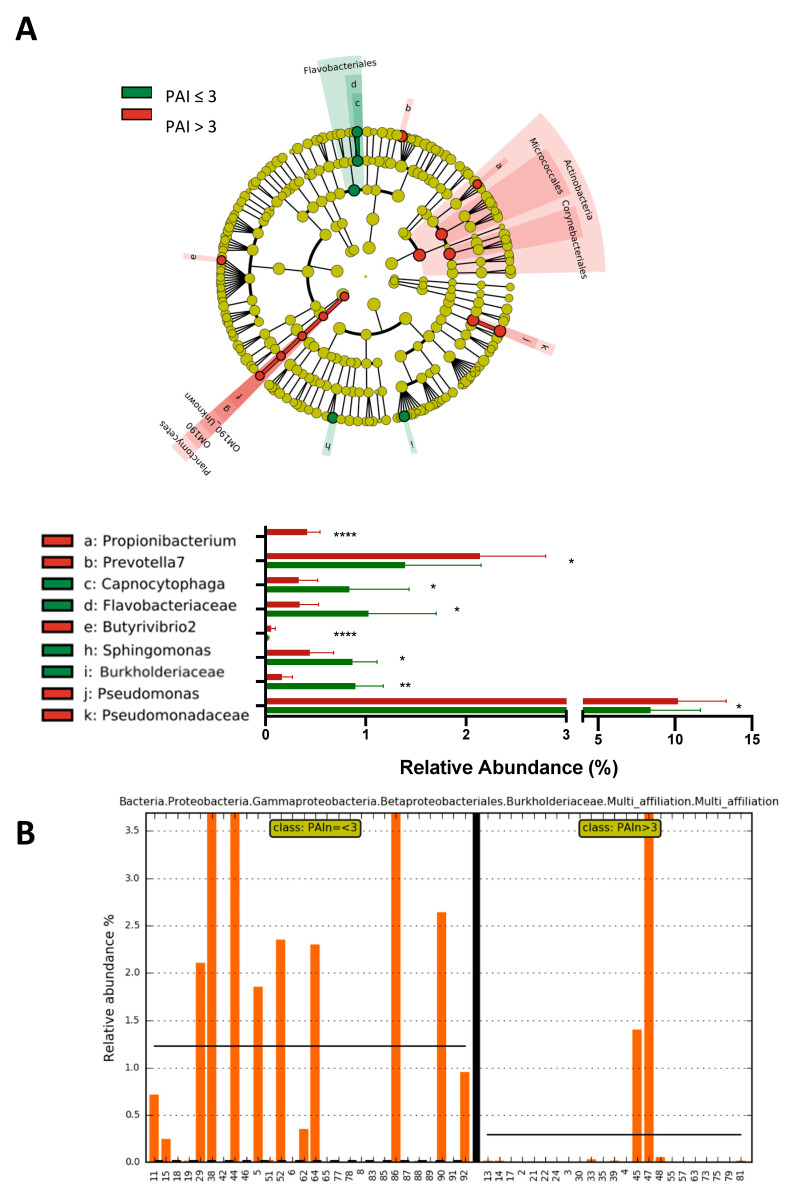
Microbiota profiles from granuloma in PAI ≤ 3 group (n = 44) show an increase in alpha diversity associated with the family Burkholderiaceae. (**A**) Linear discriminant analysis (LDA) effect size (LEfSe) cladogram of pairwise analysis for 16S rDNA sequence analysis of granuloma samples (the cladogram shows the taxonomic levels represented by rings with phyla at the innermost ring and genera at the outermost ring, and each circle is a member within that level) between two groups’–PAI ≤ 3 and PAI > 3—histogram for relative abundance (%) for taxonomic groups identified by single letters in cladogram. (Data as mean±SEM; * *p* < 0.05, ** *p* < 0.01,**** *p* < 0.0001, unpaired Mann–Whitney test and LDA score 2.0 for cladograms). (**B**) Relative abundance of the Burkholderiaceae family per sample in PAI ≤ 3 and PAI > 3 groups.

**Figure 4 ijms-24-01589-f004:**
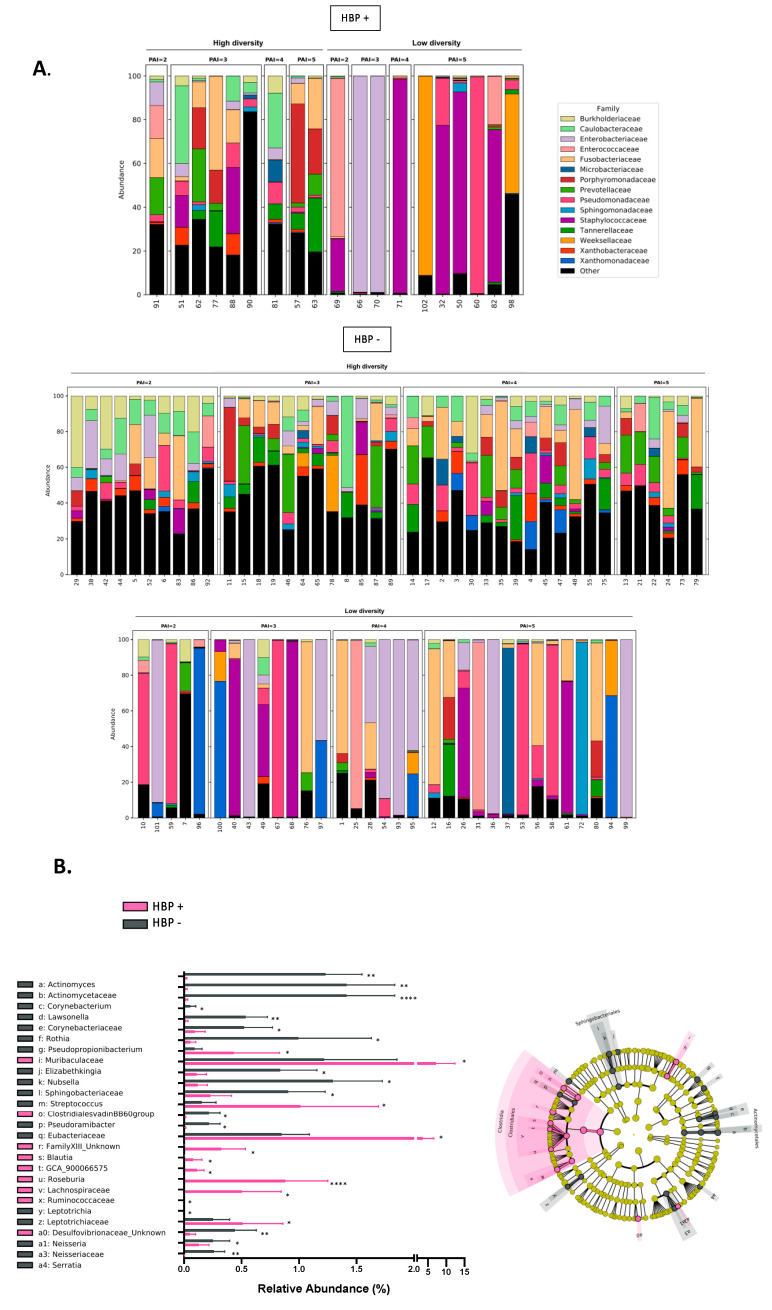
Microbiota profiles of high-diversity samples from granuloma split into the two groups: subjects with high blood pressure (HBP+) and without high blood pressure (HBP−). (**A**) Granuloma microbiota composition at family level for high diversity split into the two groups, HBP+ and HBP−. Only the top 15 most abundant families are displayed; the remaining families are aggregated in the “other” category. (**B**) Linear discriminant analysis (LDA) effect size (LEfSe) cladogram of pairwise analysis for 16S rDNA sequence analysis of granuloma samples (the cladogram shows the taxonomic levels represented by rings with phyla at the innermost ring and genera at the outermost ring, and each circle is a member within that level) between the two groups, HBP+ and HBP− The histogram for relative abundance (%) for taxonomic groups identified by single letters in cladogram. (Data as mean ± SEM; * *p* < 0.05, ** *p* < 0.01, **** *p* < 0.0001, unpaired Mann–Whitney test and LDA score 2.0 for cladograms). (**C**) Alpha diversity analyses at OTU level confirm HBP+ and HBP− microbiome profiles in granuloma. Boxplots show the distribution of bacterial richness (observed OTUs and diversity (Shannon and Simpson) values of granuloma microbiota HBP+ and HBP− diversity profiles. Statistically significant differences were determined using a Wilcoxon test (* *p* < 0.05).

**Table 1 ijms-24-01589-t001:** Clinical and oral parameters based on the periapical index score (data as mean ± SD. Unpaired Mann–Whitney test).

	All Subjects (n = 94)	PAI ≤ 3 (n = 44)	PAI > 3 (n = 50)	*p* Value
Age (years)	54.53 ± 14.22	55.40 ± 14.48	53.76 ± 13.54	*p* = 0.36
Weight (kg)	71.39 ± 13.89	73.52 ± 15.63	69.31 ± 11.99	*p* = 0.12
Height (cm)	169.67 ± 13.22	171.13 ± 07.92	169.35 ± 16.36	*p* = 0.4
BMI (kg/m^2^)	25.66 ± 13.16	24.92 ± 4.03	23.71 ± 2.84	*p* = 0.20
Stress Score (Scale 0 to 10)	4.68 ± 2.68	4.71 ± 2.85	4.56 ± 2.52	*p* = 0.55
DMF index (decayed, missing and filled)	14.63 ± 5.38	15.25 ± 5.83	14.10 ± 4.95	*p* = 0.31
Number of decayed teeth (D)	0.51 ± 1.00	0.60 ± 1.10	0.42 ± 0.90	*p* = 0.36
Number of missing teeth (M)	5.62 ± 3.86	6.18 ± 4.37	8.96 ± 3.48	*p* = 0.18
Number of filled teeth (F)	8.80 ± 4.02	8.63 ± 4.60	5.14 ± 3.32	*p* = 0.63
Number of dental brushings/day	2.01 ± 0.59	2.045 ± 0,52	1.98 ± 0,65	*p* = 0.60
PAI score	3.64 ± 1.09	2.63 ± 0.48	4.56 ± 0.50	*p* < 0.0001

**Table 2 ijms-24-01589-t002:** Clinical and oral parameters based on the diagnosis of high blood pressure (data as mean ± SD. unpaired Mann–Whitney test).

	HBP− (n = 75)	HBP+ (n = 19)	*p* Value
Age (years)	53.66 ± 13.96	52.5 ± 13.95	*p* = 0.77
Weight (kg)	71.49 ± 14.41	69.00 ± 06.63	*p* = 0.50
Height (cm)	169.09 ± 15.29	168.94 ± 5.22	*p* = 0.97
BMI (kg/m^2^)	26.29 ± 15.77	24.23 ± 2.62	*p* = 0.60
Stress Score (scale 0 to 10)	4.66 ± 2.59	4.65 ± 2.65	*p* = 0.72
DMF index (decayed, missing and filled)	14.74 ± 5.26	13.56 ± 6.03	*p* = 0.44
Number of decayed teeth (D)	0.59 ± 1.08	0.38 ± 0.89	*p* = 0.47
Number of missing teeth (M)	5.62 ± 3.77	650 ± 4.83	*p* = 3.77
Number of filled teeth (F)	8.75 ± 4.07	7.31 ± 3.55	*p* = 0.19
Number of dental brushings/day	1.98 ± 0,58	2.10 ± 0.65	*p* = 0.4437
PAI score	3.47 ± 1.05	4.05 ± 0.96	*p* = 0.0416
Diagnostic of severe periapical diseases (PAI = 5)	27 % (n = 20)	42% (n = 8)	

## Data Availability

Data is unavailable due to privacy or ethical restrictions.

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
