# Peer review of "Low-Diversity Microbiota in Apical Periodontitis and High Blood Pressure Are Signatures of the Severity of Apical Lesions in Humans"

_ijms, 2023, doi:10.3390/ijms24021589_

Round 1

Reviewer 1 Report

The authors have performed a clinical study in a hospital setting collecting patient samples for sequencing the 16SrRNA gene of AP microbiota. They have concluded that the low diversity of microbiota is associated with the highest severity of PAI. Interestingly higher Blood Pressure is found in patients with high PAI scores. Overall the study was found to be interesting. However, the authors need to explain the results better;

The hypothesis of the study design is not clear.

Higher Blood Pressure is found in patients with high PAI scores reported in this study but it is already known as per previous reports.

Authors in the discussion talked about their future work in an elaborate manner (Line 362-376). It looks like a project proposal!

Results got from the present study are not presented well or not clearly written by the authors.

Is it smocking or smoking?

Figures don't have clear legends for abbreviations.

Author Response

RESPONSE TO REVIEWERS - " Low diversity microbiota in apical periodontitis and high blood pressure are signatures of the severity of apical lesions in human" Minty etal. ijms-2132546

We greatly appreciate the general positive attitude and all of the suggestions proposed by the reviewer to ameliorate our manuscript. We really hope that we have met all of your points now

Reviewer 1

  • The hypothesis of the study design is not clear.

We thank the reviewer for this comment. To clarify the hypothesis, we have changed the introduction section line 66-69: « In fact, as support of our hypothesis, we previously described that the translocation of bacteria from the oral microbiota to inside the tooth, then to the peri-apex area and further to systemic circulation causes inflammation and the severity of the granuloma”.

« We hypothesized that the translocation of bacteria from the oral microbiota to inside the tooth, then to the peri-apex area and further to systemic circulation could induce inflammation and contribute to the severity of the granuloma »

  • Higher Blood Pressure is found in patients with high PAI scores reported in this study but it is already known as per previous reports.

Thank you for your interest. To clarify the text, we added the following sentence at the end of introduction section line 82-84 “As per previous reports, we confirm that High Blood Pressure is a frequent comorbidity in patients with high PAI scores”. Following this article https://www.ncbi.nlm.nih.gov/pmc/articles/PMC7600401/ - Relationship between Apical Periodontitis and Metabolic Syndrome and Cardiovascular Events: A Cross-Sectional Study

  • Authors in the discussion talked about their future work in an elaborate manner (Line 362-376). It looks like a project proposal!

We thank the reviewer for this precious suggestion. We agree with this remark and so we clearly reduce this part of the discussion line 368-373 ;

“To understand the corresponding molecular mechanisms of the virulence of microbiome to host immune defence cross talk, we will establish a validation it will be interesting to realize Shot gun sequence the full microbiota of the granuloma (GranulOmics) and the saliva (SalivOmics) from the discovery library. Transcriptomics will be analysed from the Granuloma and metabolomics from the saliva to identify taxa at the bacterial etiology of the AP and host molecular targets of the microbiome molecules. A precise attention will be provided to cells from the immune system. Identifying molecular pathways of the host to microbiome crosstalk associated with the healing index is planned to be our next analysis is could be interesting to study. A special regard on the immune system will be provided to identify potential immune impairments allowing the aggressiveness of the microbiota leading to the severity of AP. Eventually, CulturOmics can also be performed to isolate the bacterial candidates and in vivo animal models to validate the causal taxa.  “

  • Results got from the present study are not presented well or not clearly written by the authors.

We thank the reviewer for this comment and we decided to modify the global presentation and rewritten many parts of the results, especially  line  120 : We explored the potential link between the severity of the lesion and general health in order to identify potential clinical factors associated with the PAI score. Different clinical parameters were first measured, then subsequently subjected to principal component analysis (PCA), as shown in Figure 1. PCA aims to identify clinical risk factors potentially associated with the severity of the disease, suggesting ways in which hypertension and peri apical disease may be associated.

And line 152 : “The Enterobacteriaceae family represented more than 90% of the total abundance in 2 of 20 patients with a PAI equal to 5 and is the predominant bacterial family in 23% of subjects with a low diversity profile (Figure 2.A). The Pseudomonadaceae family represented more than 90% of the total abundance in 3 of 20 patients with a PAI equal to 5 and is the predominant bacterial family in 14% of subjects with a low diversity profile (Figure 2.A). The Staphylococcaceae family represented more than 75% of the total abundance in 5 of 20 patients with a PAI equal to 5 and is the predominant bacterial family in 21% of subjects with a low diversity profile. The Enterobacteriaceae, Pseudomonadaceae and Staphylococcaceae families are the main bacterial taxa represented in subjects with the low diversity profile for AP microbiota. The data suggested that the low diversity microbiota could contribute to the severity of PAI< 3 groups.”

  • Is it smocking or smoking?

We thank the reviewer for this remark. The right word is “smoking” and we corrected in the figures (Figures 1 A&B).

  • Figures don't have clear legends for abbreviations.

We thank the reviewer for this observation. We clarify the abbreviations figures 1, 2, 4 and the legend of the table 2.

Reviewer 2 Report

Dear Authors,  

I think that the article is well-written with a very detailed description of your study. I only suggest to modify the position of material and method section moving it before discussion. Therefore the order of every single section should be: introduction, material and methods, results, discussion and conclusion. 

Good luck for your publication. 

Best regard

Author Response

Toulouse, January 5th 2023

RESPONSE TO REVIEWERS - " Low diversity microbiota in apical periodontitis and high blood pressure are signatures of the severity of apical lesions in human" Minty etal. ijms-2132546

We greatly appreciate the general positive attitude and all of the suggestions proposed by the reviewer to ameliorate our manuscript. We really hope that we have met all of your points now.

Best regards

Reviewer 2

  • I only suggest to modify the position of material and method section moving it before discussion.

We are sorry but we followed the template given by the journal. Thank you for your positive comment on our manuscript.
